# CIR-Based Device-Free People Counting via UWB Signals

**DOI:** 10.3390/s21093296

**Published:** 2021-05-10

**Authors:** Mauro De Sanctis, Aleandro Conte, Tommaso Rossi, Simone Di Domenico, Ernestina Cianca

**Affiliations:** Department of Electronics Engineering, University of Rome “Tor Vergata”, 00133 Roma, Italy; mauro.de.sanctis@uniroma2.it (M.D.S.); aleandroconte.ing@gmail.com (A.C.); tommaso.rossi@uniroma2.it (T.R.); simone.didomenico@uniroma2.it (S.D.D.)

**Keywords:** RF sensing, UWB, people counting, channel impulse response

## Abstract

The outbreak of COVID-19 has resulted in many different policies being adopted across the world to reduce the spread of the virus. These policies include wearing surgical masks, hand hygiene practices, increased social distancing and full country-wide lockdown. Specifically, social distancing involves keeping a certain distance from others and avoiding gathering together in large groups. Automatic crowd density estimation is a technological solution that could help in guaranteeing social distancing by reducing the probability that two persons in a public area come in close proximity to each other while moving around. This paper proposes a novel low complexity RF sensing system for automatic people counting based on low cost UWB transceivers. The proposed system is based on an ordinary classifier that exploits features extracted from the channel impulse response of UWB communication signals. Specifically, features are extracted from the sorted list of singular values obtained from the singular value decomposition applied to the matrix of the channel impulse response vector differences. Experimental results achieved in two different environments show that the proposed system is a promising candidate for future automatic crowd density monitoring systems.

## 1. Introduction

Social distancing is an effective approach to limit disease diffusion [1]. During the ongoing COVID-19 pandemic, several social distancing measures have been implemented by governments: travel restrictions, border control, closing public places and keeping a 1.5–2 m distance from each other outside. In such a context, there is a growing need for technologies that are able to facilitate social distancing in different scenarios and types of environment. Several recent papers have addressed this topic [2,3,4]. One of such technological solutions is the real-time automatic estimation of the number of people inside a room, which could be used to alert when this number cannot guarantee social distancing. Such systems could be effectively deployed in museums, exhibition areas, commercial centers but also public spaces such as universities. Several approaches for automatic crowd-counting have been proposed in the literature. Two main classes of people counting technologies can be identified [5]: device-based or device-free [6]. The first class estimates the number of people inside an environment by analyzing Radio Frequency (RF) signals transmitted by sensors carried by the monitored people, such as Radio-frequency identification (RFID) tags or smartphones [7]. Among device-based solutions, much interest has been recently devoted to counting systems that correlate the number of people to the number of connections from user devices (e.g., smartphones) to WiFi access points [8] or through passive sensing of WiFi probe request signals [9], periodically transmitted by monitored people devices (without the need for direct connection to access points). Such approaches have some major drawbacks; the first is the need for the monitored people to carry mobile devices or specific sensors (in this latter case the monitored person has to be cooperative). Moreover, for the WiFi based approaches, there can be the dependency on the direct connection to the access point, the need for a switched-on WiFi interface on monitored device and the fact that the performance can be biased by the number of devices carried by each monitored person. Furthermore, new MAC address randomization techniques does not easily allow identification of Probe Request packets sent by different devices/users.

In the device-free approach, there is no need for people to carry any device. The most traditional device-free approach is based on video image processing [10] but this has several limitations and drawbacks such as being prone to occlusion, high deployment costs and concerns related to privacy.

The alternative device-free approach is based on the use of RF signals that are already present in the monitored environment (opportunistic signals) or specifically generated for people counting purposes. This technology is based on the fact that RF propagation channel is continuously modified by the presence of people. This in turns means that it is possible to analyse RF signals to identify changes produced by the presence of people and extract information on the number of persons inside the monitored environment. Most of these device-free approaches are based on WiFi signals [11,12] but some recent works have proved also the possibility to use Long Term Evolution (LTE) signals for radio analytics applications [13,14]. The advantage of using either WiFi or LTE signals is the possibility to reuse part of the existing infrastructure (transmitters are already deployed). Moreover, in case of WiFi, commercial low cost devices which allow to extract the Channel State Information (CSI) can be used for collecting the data to be processed. Higher accuracy is expected by solutions that use higher bandwidth signals, such as Ultra-Wideband (UWB) signals [6]. As a matter of fact, several crowd counting systems using UWB have been proposed [6,15,16,17,18]. Most of them are based on a radar-approach, counting each signal separately and assuming that different clusters of signals are reflected by the persons in the monitored area. This paper shows the feasibility of a novel non-radar approach, based on commercial low cost UWB transceivers, from which it is possible to extract the Channel Impulse Response (CIR). The Singular Value Decomposition (SVD) is used to identify independent patterns in the time variation of the CIR. The paper shows that these independent patterns are correlated to the number of people inside the room. Therefore, specific features are extracted from the measured CIR and used to train a classifier. This paper shows the feasibility and effectiveness of this novel approach in two different environments. The number of people considered is up to the maximum number of people allowed in the room considering the COVID-19 restrictions: 5 persons in a 3×4.5
$m^{2}$ room and 8 persons in a 5×5
$m^{2}$ room. Results are very encouraging also considering that the used transceivers have a bandwidth of 500 MHz, which is lower than the bandwidth, namely 2 GHz of radar-devices commonly considered for crowd counting systems. The proposed approach allows to fully take advantage of the wide bandwidth of UWB in a multipath-rich environment. This paper is organized as follows: Section 2 presents an overview of UWB-based counting systems; Section 3 describes the chosen features extracted from the CIR; Section 4 presents the experimental set-up and Section 5 presents the results; conclusions are drawn in Section 6

## 2. Related Works and Contribution

Among radio-based device-free counting systems, several radio signals have been considered: WiFi [19,20,21], LTE [22,23,24,25] and UWB [15,16,17]. In the case of WiFi or LTE systems, counting is performed by using some correlation between the number of people in the room and some features extracted by the received signal, in particular from the Channel State Information (CSI). Many of the proposed WiFi-based counting systems have the practical advantage of using low cost commercial transceivers from which is possible to extract the CSI of the received signal thus enabling a lost cost counting system infrastructure. On the other hand, most of the UWB-based proposed counting systems use a radar-approach. Many initial works on UWB-based counting systems are based on multi-target radars where the number of targets is deduced by the set of measurements associated with each detected target, for example its estimated position. More recent works have focused on the so-called crowd-centric methods [6], where targets number is estimated without their position. In [16], to find the *patterns* correlated to the number of people, major clusters from the signal are detected and the amplitudes of main pulses having the maximum amplitude among the pulses constituting each cluster are found. Such radar-based systems have the following limitations: (i) they require Line-of-Sight (LOS) for most of the time between the radar sensor and the targets to be counted and this is not possible in all the scenarios, especially in dense people scenarios or environments with a not *regular* shape and much furniture in it; (ii) multipath is a problem to be faced and most of the processing effort is on the removal of the multipath components not directly related to the people to be counted. In [15], a solution for dense people scenario has been proposed. The authors of [6] propose a crowd-centric approach for counting that relies on energy detection. Moreover, in [6], a tractable theoretical model is provided for describing the relation between the number of target and the energy samples. This paper proposes a novel not-radar approach using low cost commercial devices from which it is possible to extract the CIR. Low cost UWB transceivers have been also considered for people counting in [17], but for a very simple scenario where TX and RX are placed on the doorway and CIR is analyzed to detect the obstruction of the LOS due to the people passing through. This is the first work proposing the use of such low cost transceivers for people counting in a more general scenario where people move randomly inside a room. The idea is that when people move inside the room, they cause changes in the propagation channel according to some patterns that can be considered statistically independent. In our proposed approach, these *patterns* can be detectable in the received signal even if the received multipath components is not directly reflected from the person, but they reach the receivers through different paths. This makes the proposed system less dependent from the specific environment and deployment (position of TX and RX with respect to the area in which people move).

## 3. Proposed Counting System and Feature Extraction

The proposed crowd density estimation system is based on the idea that when people move inside the room, they cause changes in the propagation channel according to some patterns that can be considered statistically independent. In previous works, working on the same principle [14,19], where the LTE or WiFi radio signals are used for counting, the temporal variation of the channel response is *observed* in the frequency domain by collecting and analyzing the so-called CSI. In this work, we are using UWB signals that are characterized by much higher bandwidth: in our experimental set-up, the bandwidth is 500 MHz, while in [14] the bandwidth of the LTE signal is 15 MHz. Therefore, we have chosen to work in the time domain, using the CIR rather than the CSI. We expect that by analysing the variation in time of the CIR, it is possible to identify a number of independent patterns of temporal variations that is correlated to the number of people in the room. The number of large singular values of the matrix containing the collected CIRs over a given time window, is expected to be proportional to the number of independent patterns and hence the number of people in the room. Therefore, we perform the SVD on the matrix having the CIR vector differences on the rows and we have chosen features that characterize the singular values trend.

Let us denote with ${\hat{h}}_{k}$ the complex-valued CIR vector estimated at time *k* and let us denote with ${\hat{a}}_{k}$ the vector containing the magnitude of the elements of ${\hat{h}}_{k}$. Consider the set $A=\{{\hat{a}}_{l+1},{\hat{a}}_{l+2},\ldots ,{\hat{a}}_{l+W}\}$ of W=100 consecutive CIR magnitude vectors, corresponding to 30 seconds of channel measurements. We first compute the matrix D of CIR magnitude vector differences, taking all vector differences of pair of vectors in A. As a consequence, D has $\frac{W(W-1)}{2}$ rows. The next step is to apply the SVD to the matrix D resulting in the factorization of D into the product of three matrices as follows:(1)$$D=U\Sigma V^{T},$$
where the columns of U are eigenvectors of ${\mathrm{DD}}^{T}$, and the columns of V are eigenvectors of $D^{T}D$. Σ is a pseudo-diagonal matrix and the *r* singular values $s_{n},n=1,2,\ldots ,r$ on the diagonal of Σ are the square roots of the nonzero eigenvalues of both ${\mathrm{DD}}^{T}$ and $D^{T}D$.

The singular values are usually sorted in descending order. The reason to sort the list of singular values follows from the following physical motivation. Specifically, relatively large singular values are associated with independent patterns that can be identified in the data matrix D while relatively small singular values are associated to the noise of the data matrix D. Assuming that variations of the CIR vectors induced by the motion of different persons are independent, we expect that the value and the number of the largest singular values increase as the number of people increases.

Therefore, considering the curve built using the singular values ordered in descending order (monotonically decreasing curve), we expect that this curve moves up as long as the number of people increases. This intuition is confirmed by Figure 1, where the curve of singular values is shown for the data matrix D collected in a 3×4.5
$m^{2}$ room with different number of people; the ordinate reports the singular values while the abscissa reports the index of the singular value. Therefore, a feature extraction method based on the characterization of this curve built on the singular values of CIR complex magnitude vector differences D is expected to be a good choice for the classification process.

In particular, with reference to Figure 1, let us denote with y(x) the curve built using the singular value sequence, $y_{n}=y\left(x_{n}\right)$, where $x_{n}=\left\{x_{1},x_{2},...,x_{r}\right\}$ are the singular value indexes. We define the following set of features.

The definition of the *m*-th slope of y(x) is provided through the following Equation:(2)$$f_{s}^{m}=\frac{y_{m}-y_{m+1}}{x_{m}-x_{m+1}}.$$

The definition of the average slope of y(x) is provided through the following Equation:(3)$$f_{s}=\frac{1}{r}\sum_{m=1}^{r}f_{s}^{m},$$
where *r* is the number of points of the curve.

The definition of the center of gravity of y(x) is provided through the following Equation:(4)$$f_{c}=\frac{\sum_{i=1}^{r}y_{i}x_{i}}{\sum_{i=1}^{r}y_{i}}.$$

The definition of the area under the curve y(x) is provided through the following Equation:(5)$$f_{a}=\sum_{i=1}^{r-1}y_{i}\cdot \left(x_{i+1}-x_{i}\right).$$

## 4. Experimental Set-Up

We have conducted experiments in two different rooms:Room A: it is a 3×4.5
$m^{2}$ room, where furniture is mainly placed close to the walls and people can move rather freely;Room B: it is a 5×5
$m^{2}$ room, where furniture is placed also in the middle of the room and people movements are more constrained.

In Figure 2, the layout of the two rooms is shown. The light blue rectangles represent desks, the dark blue rectangles represent shelves of about 2 m in height and the lines represent PC desktops placed on the desks. Two Decawave DWM1001 UWB modules working as transmitter and receiver have been placed on two desks at the opposite sides of the rooms (see Figure 2). The distance between TX and RX was between 4 to 5 m. The signal bandwidth is 500 MHz and the transmission is set to one pulse each 300 ms. The receiver stores a vector of 1000 samples of the CIR, with a sampling period of 1.016 ns. We have used J-Link Commander to connect the target RX module to a computer, to monitor its status, to halt it and to begin the frames’ reception and J-Link RTT viewer to print the log of the CIR amplitudes. In both experiments, volunteers could move randomly inside the room or stand firm. The CIR vectors have been collected for five minutes and for each group of people in the two rooms, ranging from 0 to 5 people in room A and from 0 to 8 people in room B. The collected CIR data are processed offline to build the dataset, performing data fusion, feature extraction, feature selection and classification tasks.

Figure 3 and Figure 4 show the normalized amplitude of the CIR measured in room A when the room is empty and when 5 persons are moving around, respectively. In those Figures, CIRs have been averaged over the captured frames (i.e., 1768) and plotted from sample 740 to sample 800 of the 1000-samples long vector extracted from the device. It is worth highlighting that before sample 740, CIR amplitudes stored in the extracted CIR vector are zero. Moreover, from Figure 3 and Figure 4, it is evident that already before sample 800, there are no more significant signal replicas and the CIR vector contains mainly noise. Therefore, in the following, we work with 60 samples long CIR vectors (from sample 740 to sample 800). Moreover, from Figure 3 and Figure 4, it can be also observed that when the room is empty, there are few clear peaks associated with the main *scatterers* of the room (i.e., walls, tables, chairs). When people are present in the room as in Figure 4, the curve is smoother as the variance of the CIR over time is larger due to the movements of some of the *scatterers* (i.e., people).

## 5. Experimental Results

In the following, we show that results with two different types of classifiers: Naïve Bayes classifier, Decision Tree [26]. After the computation of features for each different number of people, the classifier is applied to every possible combination of features from the overall set of 5 features previously defined: 1-st slope, 2-nd slope, average slope, center of gravity, area under the curve. Then, an exhaustive search was carried out to find the best set of features that maximizes the average accuracy of the classification. As a result, we found that the best combination of features is the triplet of features: 2-nd slope, center of gravity and area under curve. Scatter plots of the pair of features 2-nd slope and center of gravity are shown in Figure 5 and Figure 6. In case of room B, assuming acceptable an error of ±1 person, we have considered the following classes: empty, 1 person, 2 person, 3–4 persons, 5–6 persons, 7–8 persons. Features are extracted using a sliding window mechanism with a window size of W=200 CIR vectors corresponding to 60 s. Table 1, Table 2, Table 3 and Table 4 show the confusion matrices for the two rooms in case of Naïve Bayes classifier and Decision Tree. First of all, it can be noticed that in both rooms, the Decision Tree classifier provides better performance: in room A the Decision Tree classifier provides an average accuracy of 99% with respect to 96% of the Naïve Bayes classifier. Accepting an error of ±1, the average accuracy is in both cases 100%. It has been also verified that by reducing the sliding window size from 60 s to 30 s, the accuracy in room A drops slightly to 96% in case of Decision Tree and 93% for the Naïve Bayes classifier, which still represents a very good value of accuracy. In room B, accepting an error of ±1, the average accuracy is 89% with the Decision Tree and 86% with Naïve Bayes. The worse performance in room B is expected as the room B has a more complex layout, with furniture and equipment (tables, monitors of PC etc) that did not guarantee LOS propagation conditions between the target and the receiver for most of the time. Therefore, the proposed counting system is expected to work properly in different environments without constraints, usually present in other radar-based counting systems, on the *angular* view of the counting system, the position of the TX and the position of furniture inside the room.

As benchmark, Table 5 reports the performance of other passive device-free RF-based crowd counting methods, using either WiFI, LTE or UWB signal. Even if a fair comparison between different methods is not possible as performance is influenced by several aspects of the experimental setup (signal bandwidth, carrier frequency, distance between transmitter and receiver, number of transmitters and receivers, maximum number of people, room size, constraints on the position of TX and RX etc.), it is possible to state that the achieved results are very encouraging, considering that the used transceivers have a bandwidth of 500 MHz, which is lower than the bandwidth, namely 2 GHz, of radar-devices commonly considered for UWB-based crowd counting systems. The proposed approach allows to fully take advantage of the wide bandwidth of UWB in a multipath-rich environment. It is worth outlining that the same approach and processing could be used with other opportunistic radio signals, even characterized by lower bandwidth. In particular, we also evaluated the performance when lower bandwidth signals are available. For a bandwidth of 250 MHz, in room A we have got an average accuracy of 93% and for a bandwidth of 125 MHz we have got an accuracy of 88%, which are still good results when compared with other low cost device-free passive counting systems.

## 6. Conclusions

This paper presents a novel UWB-based people counting system, which uses the CIR extracted from low cost commercial UWB transceivers, more commonly used for localization purposes. The proposed system has been tested in two different environments and the achieved results (100% accuracy up to 5 people and 89% up to 8 people) are very promising and encourage further investigations of its performance for a much higher number of people and also considering a larger variety of application environments. We could not perform such experiments due to COVID-19 limitations on the number of people allowed in one room. Moreover, the proposed approach, which works on the differences between two consecutive CIR, is expected to be rather independent from the background environment, thus making the proposed counting system independent from the room where the training is performed. Therefore, the training phase could be performed once in a *standard* room and then the system could be implemented in different rooms. This will be the subject of further investigations.

## Figures and Tables

**Figure 1 sensors-21-03296-f001:**
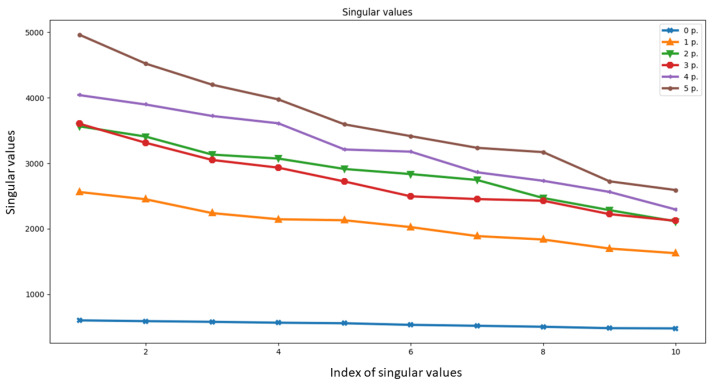
Singular values for different number of people in room A.

**Figure 2 sensors-21-03296-f002:**
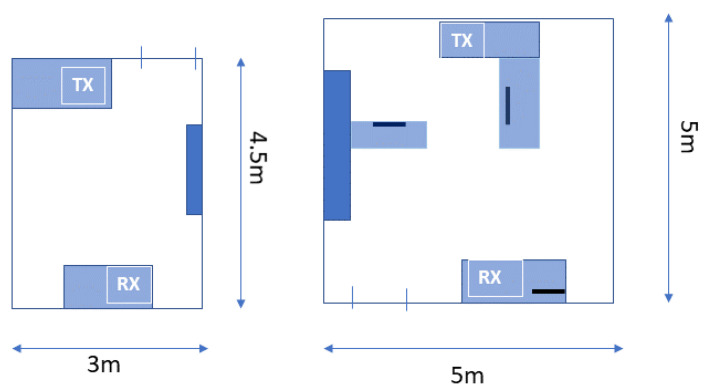
Layout of Room A and Room B.

**Figure 3 sensors-21-03296-f003:**
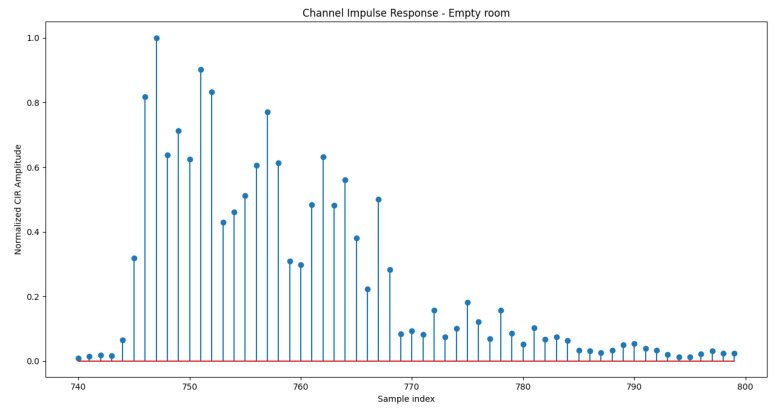
Normalized Channel Impulse Response, Room A, empty.

**Figure 4 sensors-21-03296-f004:**
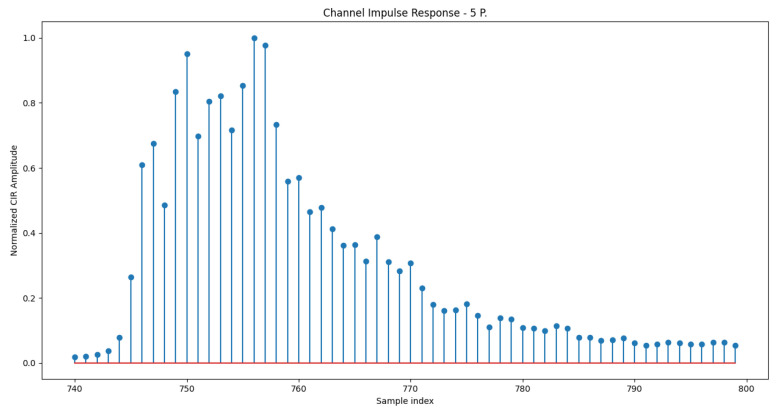
Normalized Channel Impulse Response, Room A with 5 People moving in the room.

**Figure 5 sensors-21-03296-f005:**
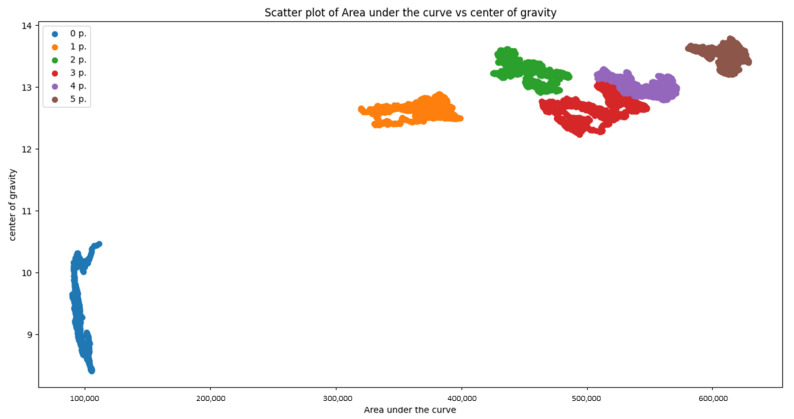
Scatter plot of the features *Area under the curve* vs. *Center of gravity*-for the different classes in Room A with maximum 5 people.

**Figure 6 sensors-21-03296-f006:**
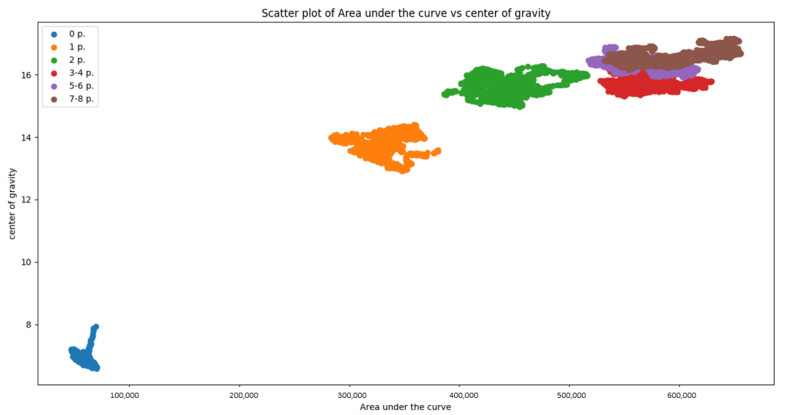
Scatter plot of the features *Area under the curve* vs. *Center of gravity*-for the different classes in Room B with maximum 8 people.

**Table 1 sensors-21-03296-t001:** Confusion matrix in room A with maximum 5 people and Naïve Bayes classifier.

Predicted						
	**Empty**	**1**	**2**	**3**	**4**	**5**
**empty**	**1**	0	0	0	0	0
**1**	0	**1**	0	0	0	0
**2**	0	0	**1**	0	0	0
**3**	0	0	0	**1**	0	0
**4**	0	0	0	0	**0.94**	0.06
**5**	0	0	0	0	0.17	**0.83**

**Table 2 sensors-21-03296-t002:** Confusion matrix in room A with maximum 5 people and Decision Tree Classifier.

Predicted						
	**Empty**	**1**	**2**	**3**	**4**	**5**
**empty**	**1**	0	0	0	0	0
**1**	0	**1**	0	0	0	0
**2**	0	0	**1**	0	0	0
**3**	0	0	0	**0.98**	0.02	0
**4**	0	0	0	0.04	**0.96**	0
**5**	0	0	0	0	0	**1**

**Table 3 sensors-21-03296-t003:** Confusion matrix in room B with maximum 8 people and Naïve Bayes classifier.

Predicted						
	**Empty**	**1**	**2**	**3–4**	**5–6**	**7–8**
**empty**	**1**	0	0	0	0	0
**1**	0	**1**	0	0	0	0
**2**	0	0	**1**	0	0	0
**3**	0	0	0	**0.79**	0.13	0.08
**4**	0	0	0	0.26	**0.74**	0
**5**	0	0	0	0	0.36	**0.64**

**Table 4 sensors-21-03296-t004:** Confusion matrix in room B with maximum 8 people and Decision Tree classifier.

Predicted						
	**Empty**	**1**	**2**	**3-4**	**5-6**	**7-8**
**empty**	**1**	0	0	0	0	0
**1**	0	**1**	0	0	0	0
**2**	0	0	**1**	0	0	0
**3**	0	0	0	**0.86**	0.1	0.03
**4**	0	0	0	0.11	**0.69**	0.2
**5**	0	0	0	0.02	0.18	**0.79**

**Table 5 sensors-21-03296-t005:** Comparison of device-free crowd counting methods.

Ref	Room Size	Max # of People	Signal	Accuracy
[21]	33 m${}^{2}$	9	WiFi	P(e ≤ 2) = 63%
[27]	-	7	WiFi	94%,(0p, 1p–3p, 4p–7p)
[28]	30 m${}^{2}$, 70 m${}^{2}$	7	WiFi	73%, 63% (0p, 1p, 2p, 3p–4p, 5p–7p)
[14]	45 m${}^{2}$	17	LTE	84% (0p, 1p, 2p–4p, 5p–12p, 13p–17p)
[15]	45 m${}^{2}$	15	IR-UWB	98.7%
[16]	25 m${}^{2}$	10	IR-UWB	P(e ≤ 1) = 93.6%

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
