# Peer review of "CIR-Based Device-Free People Counting via UWB Signals"

_sensors, 2021, doi:10.3390/s21093296_

Round 1

Reviewer 1 Report

Through the literature review, feature extraction and the conclusions part of this article, it can be seen that you have read a lot of literature and studied in-depth how indoor UWB determines the number of people. Moreover, the principle is clear, and the literature is well-organized.

However, there are still some problems in this article. First, the first part and the second part explain a large amount of literature, there are some repetitions, and can be integrated into a literature review to lead to your own improvement method. And the third paragraph of the first part is too long. According to the general opinion, pay attention to the division of paragraphs, and the length of the paragraph should be appropriate. In addition, there are many abbreviations that appear for the first time, such as CSI on line 58, LOS on line 65, and TX and RX on line 74. And in line 61, two most appear in a row. These details need to be paid attention to.

Second, the second part of the content is not suitable here, you can integrate this part of the content into the literature review as mentioned before. In the second part, you can introduce the principle of how to use UWB to determine the number of people in the room. The basis of feature selection, Bayesian theory, classifier training, in the experimental part, can be placed in this part. It is best to integrate these theoretical parts and the third part of the theory into another two chapters.

Third, since the experimental environment is explained in Chapter 4, it is no longer needed in the third paragraph of the introduction. And when introducing the indoor environment, it is best to attach a schematic diagram to facilitate readers' understanding. And pay attention to the layout and content to be compact. For example, Figure 2 and Figure 3 appear on the fifth page. It is best to follow the chart immediately on the sixth page, rather than appearing at the bottom of the sixth page and the seventh page. Figure 1 should also list what the coordinate axis means on the coordinate axis. And there is a lack of comparative experiments in the experimental part in the aspect of accuracy.

From the principle and experimental parts, you had proposed a method of using UWB to determine the number of people in the room with good accuracy, and the training process is simple. I hope you can get rid of the above problems.

Looking forward to your changes.

Author Response

We thank the reviewer for the useful suggestions to improve the paper. We have modified the paper to follow such suggestions. In particular:

  • We have shortened the introduction and move some parts in the next Sections. The introduction was actually redundant.
  • We have checked the acronyms
  • We have attached a figure with the layout of the tested rooms
  • We have moved the figures and tables closely to the part of the text where they are referred.
  • We have introduced a table reporting the performance of other literature counting systems based on RF signal (Wifi, LTE and UWB)

Reviewer 2 Report

This paper presents a method to count the number of persons in a room by means of analyzing the Channel Impulse Response (CIR) of an Ultra-Wide-Band signal. The authors did find that the eigenvalues obtained from a vector build from 100 measures of the CIR, when sorted in decreasing order, are positively correlated with the number of persons in a closed room. The authors suggest a bayes classifier to count the number of persons in rooms. The authors have tested their method in two settings (two different smallish rooms, with 5 and 8 persons).

The idea is very nice and practical. The graph in Figure 1 shows very clearly the strength of this method. And the installation is easier than in competing approaches that, e.g., require all users to wear a device.

However, there are a few limitations in the method proposed. It is unclear how it will behave in larger rooms and more people; the authors suggest to use this system in museums, where rooms are typically huge, and a very large of attendees are expected. Hence, the paper as proposed is sufficient for a proof of concept, but does not explore the limits of the technology w.r.t. the size of the monitored space, and the number of people.

Also a limitation of this method is that it must be re-trained for each room, for which you need recordings for each possible number of persons in the room (e.g., if you want to install it in a new emplacement, and count up to 100 persons, you need 100 recordings of 1-100 persons on that setting). Also, foreseeably, calibration/training would be required every time the furniture of the  room changes. This limits the reach of this paper. 

Still, as a proof of concept is a nice idea.

The paper is easy to read and the method is well explained. Readability should be improve by:

  • Take care of introduce abbreviations before being used (e.g., CSI, UWB, LoS).
  • Reduce the overlap between introduction and related work sections.
  • The font size used in figures matches the font size of the main text.
  • Captions of Figures would include more detailed descriptions. Particularly  on Figures 4 and 5.

Author Response

We thank the reviewer for the useful suggestions and also considerations. As the reviewer said, the paper is a proof-of-concept. Reviewer mentions two challenging conditions: bigger rooms and more people in the room. When bigger rooms are considered, the main problem is that the distance from the TX and RX must be kept in a way that it possible to receive signals replicas with enough strength. This can be done by covering the room with more than one TX and RX. Therefore, the main challenge is actually the higher number of people. However, as a matter of fact, it was not possible to test the limits of the proposed solution with a higher number of people as we could not perform experiments respecting the current anti-covid rules.

About the limits related to the need of training, this is the limitation of most of the device-free RF passive systems. The proposed solution works on the difference between consecutive CIR vectors, and hence it should in principle be independent from the background environment. An important further step could be to modify the proposed solution to achieve a trained-once method, where the training is performed once in a room and could be applied to other rooms with similar characteristics. However, we have left this to further investigations, as stated in the conclusions.

About the readability:

  • Abbreviations have been checked
  • Redundancy in the introduction has been removed
  • Captions of Figures 4 and 5 has been slightly changed.

Reviewer 3 Report

Typos:
- line 155: patters -> patterns
- make LOS explicit once (line of sight)

This paper presents an original scientific work that can have important applications. The description is precise but probably a little too synthetic. The reading of the previous work cited [15] helps to understand the paper. But it gives some leads and good indications if one wants to try to explore this way.

By its interest, I am in favor of the publication of this short paper. However, I would have preferred a longer paper describing more precisely the experiments: the methods and software used, the more precise experimental conditions, the repeatability, and so on... Without going as far as a reproducible science approach that requires the publication of the code and the documentation of the experiments, all these elements would allow colleagues to more easily redo experiments. 

This is indeed discussed in the conclusion, but another point that should be explored is classification. We can improve the learning phase but we can also explore other classification algorithms. This is now facilitated by modern tools, such as scikit-learn for example, which gives a set of algorithms to facilitate this choice.

Author Response

We thank the reviewer for the useful suggestions to improve the paper.

To follow the suggestions, we have slightly modified the experimental setup section to include more details in the experiment and the steps that have been followed to get the results.

We thank the reviewer for the suggestion to test different classifiers. We have assessed three different classifiers: Naïve Bayes, NN and Decision Tree. NN provided very similar results than Naïve Bayes, while Decision Tree has provided better performance. Therefore, we have introduced also results for the Decision Tree Classifier in the revised version of the paper.